# An Egocentric Network Contact Tracing Experiment: Testing Different Procedures to Elicit Contacts and Places

**DOI:** 10.3390/ijerph18041466

**Published:** 2021-02-04

**Authors:** Andrew Pilny, C. Joseph Huber

**Affiliations:** Department of Communication, College of Communication & Information, University of Kentucky, Lexington, KY 40506, USA; Joseph.Huber@uky.edu

**Keywords:** contact tracing, ego networks, experimental design, social networks, field theory

## Abstract

Contact tracing is one of the oldest social network health interventions used to reduce the diffusion of various infectious diseases. However, some infectious diseases like COVID-19 amass at such a great scope that traditional methods of conducting contact tracing (e.g., face-to-face interviews) remain difficult to implement, pointing to the need to develop reliable and valid survey approaches. The purpose of this research is to test the effectiveness of three different egocentric survey methods for extracting contact tracing data: (1) a baseline approach, (2) a retrieval cue approach, and (3) a context-based approach. A sample of 397 college students were randomized into one condition each. They were prompted to anonymously provide contacts and populated places visited from the past four days depending on what condition they were given. After controlling for various demographic, social identity, psychological, and physiological variables, participants in the context-based condition were significantly more likely to recall more contacts (medium effect size) and places (large effect size) than the other two conditions. Theoretically, the research supports suggestions by field theory that assume network recall can be significantly improved by activating relevant activity foci. Practically, the research contributes to the development of innovative social network data collection methods for contract tracing survey instruments.

## 1. Introduction

Because infectious diseases like COVID-19 can be spread through respiratory droplets from being near and communicating with infected individuals, researchers have taken advantage of contact tracing to help mitigate the spread of the virus. Contact tracing is the process of retrospectively identifying persons who may have made interpersonal contact with a confirmed infectious individual [1]. The overall logic of contact tracing is to prevent individuals who made contact with the focal individual from further spreading the disease themselves and thus, to break the chain of infection within the larger system.

At its core, contact tracing is a form of egocentric network analysis (ENA) and, thus, one type of social network intervention to influence public health. ENA is the study of individuals (i.e., egos) and people in their immediate social environment (i.e., alters). Conducting a reliable and valid ENA study is challenging, and there is a rich academic literature dedicated to improving the rigor behind these efforts [2,3]. Indeed, the effectiveness of contact tracing is only as good as the data that can be extracted. If ideal standards are not followed for collecting contact tracing data, then most efforts to take advantage of this data will likely fall short. Potential at-risk contacts may never be discovered for treatment/quarantine and they could continue to propagate COVID-19. Likewise, hotspots and clusters are likely not to be discovered, preventing the public from being informed to follow social distancing. Moreover, recent simulation research has shown that in order for contact tracing to be effective for COVID-19, individuals must be able to successfully recall at least 80% of their contacts [4]. As such, the purpose of the proposed research is to test the effectiveness of three different egocentric survey methods for extracting contact tracing data.

### 1.1. A Survey-Based Approach for Contact Tracing

In traditional egocentric network research, face-to-face interviews are often considered the gold-standard data collection mode [3]. The reasoning primarily has to do with reduced cognitive burden for the participant. For instance, telephone interviews or video conferencing introduce auditory and visual challenges (e.g., noise quality, background activities), and survey approaches introduce challenge with participants having to correctly navigate an egocentric design without assistance, which is more complex than traditional Likert-style questions [2] that can be cognitive burdens notably absent in face-to-face interviewing.

However, there are growing challenges to implementing face-to-face interviews for contact tracing with respect to infectious diseases like COVID-19. First, face-to-face interviews are risky endeavors because the interviewer may put themselves at risk for contracting the infectious disease, especially when diagnoses specifically require a patient to quarantine. Second, interviewers need to be appropriately trained and inevitably vary in how skilled they are, resulting in interviewer effects [3]. As such, it is not surprising that there is a shortage of contact tracers appropriately trained to do such research [5]. Finally, as technology mediums continue to alter social environments [6], preferences for communication modalities must be continually monitored. For instance, in the food service industry, more people are using mobile food delivery apps on smartphones rather than calling in orders directly to restaurants [7].

These challenges to collecting contact tracing data via face-to-face interviews points towards the need to developing a reliable survey approach that could be instantly administered and standardized across various contexts. Following past egocentric network research, we examine the impact of three different survey designs: (1) baseline, (2) retrieval cue, and (3) context-based approaches. Each is described below.

### 1.2. Theory and Hypothesis Development

Perhaps the most important obstacle for egocentric data collection is recall and forgetting [3]. For the ego, contract tracing can be a cognitively intensive task. Especially when an individual has just learned that they are infected with a potentially fatal virus, it can be easy to forget nonchalant interactions from the past several days. Furthermore, data from egocentric designs rely on self-reported network interactions, which are cognitive representations that may differ from actually observed interactions (for a review, see [8]). Moreover, prior literature has suggested that successful recall (i.e., remembering all of your contacts) is influenced by a number of factors because contacts tend be cognitively stored in clusters or what Brashears and Quintane [9] call chunked substructures. Likewise, there may be individual differences in recall pertaining to demographic and social identity factors like gender, age, and race [10,11] and psychological factors like mood and well-being [12,13].

A baseline approach to surveying individuals about their contacts would be to simply ask them about who they made contact within a given period of time. This is commonly known as the interaction approach [14] because the goal is to find simple interactions rather than socially constructed relationships like friendship or individuals who provide important resources for the ego. For instance, Mossong and colleagues [15] define a contact as “EITHER a two-way conversation with three or more words in the physical presence of another person, OR physical skin-to-skin contact (for example a handshake, hug, kiss or contact sports).” Then, they ask each participant to “Write down every person that you contact during the day, regardless of whether the contact was long or short, and whether you know the person or not.” In addition to baseline approaches, other more complex methods for ENA exist as well.

The retrieval cue approach for ENA was formalized by Hsieh [10]. This approach assumes that the “successful recall of an event depends primarily on how well the retrieval cues match the event’s representations in one’s memory organization” (p. 3). A retrieval cue is any additional piece of information that helps an ego remember a past event. The retrieval cue approach mirrors Tulving’s [16] theory of cue-dependent forgetting, which assumes that forgetting, the inability to recall something in the present that could be recalled in the past, does not mean that the memory is lost, but only temporarily inaccessible. In Hsieh’s study, participants were randomly assigned to a retrieval cue condition in which they were instructed to look at their (1) cellphone contact list, (2) last 30 emails, and (3) friend list on Facebook and Twitter as retrieval cues. Hsieh’s results found that this approach yielded more contacts than a baseline approach.

Beyond contacts, it is also important for egos to remember populated places because individuals could have made contact with strangers whom they may not remember (e.g., a restaurant server). Likewise, overlapping data from multiple infected individuals who visited the same location could reveal “hotspots” that may be useful for public health purposes. As such, we hypothesize the following:

**Hypothesis** **1.**
*A retrieval cue approach will elicit more (a) contacts and (b) places visited than a baseline approach.*


The context-based name generator was formalized by Bidart and Charbonneau [17]. The context-based name generator begins by asking individuals about the social context cues of everyday life (e.g., work activity, shopping, home life). Then, once relevant contexts are triggered, contact names corresponding to those contexts are generated by the individual (see also [18]). As Bidart and Charbonneau explain, the context-based approach is motivated by field theory [19]. Field theory regards “situated action” as a key unit of analysis (i.e., action that is not stripped away from a larger context). One of the premises of field theory is that most interpersonal interaction occurs because of shared activity foci. Activity foci are simply “aspects of extra-network social structure that systematically produce patterns in networks” and include “social, psychological, legal, or physical entities around which activities are organized” [19] (p. 1016). Indeed, subsequent research has shown that priming individuals with social context can help improve network recall [11,20]. As such, we hypothesize the following:

**Hypothesis** **2.**
*A context-based approach will elicit more (a) contacts and (b) places visited than a baseline approach.*


Finally, it is important to ask about whether or not a retrieval cue or context-based approach will significantly yield more contacts and places. To our knowledge, no study has specifically compared these two approaches and there exists little theoretical basis for expecting more or fewer differences. Thus, we keep this comparison as a research question as follows:

**Research** **Question** **1.**
*How do retrieval cue or context-based approaches differ in their ability to elicit more (a) contacts and (b) places visited?*


## 2. Materials and Methods

### 2.1. Design

The experiment was a one-way between-subjects design. The related manipulation was contact elicitation, with three levels (baseline approach, retrieval cue approach, and context-based approach). Other unrelated control variables were uniformly collected across all participants (see Measures). There were two primary dependent variables: (1) unique contacts and (2) unique places visited. An a priori power analysis with the program GPower [21] suggests that at least 159 participants are needed to detect medium-sized effects (1 − β = 0.80, f = 0.25, α = 0.05).

### 2.2. Participants

To recruit participants for the experiment, we used convenience sampling from a college-wide research participation pool from a large southern university. A university sample was chosen for two reasons. First, COVID-19 restrictions (e.g., closed businesses, stay-at-home orders, etc.) differ state-by-state. In order to control for policy differences, we opted for keep geographic location constant. Second, there is increasing attention paid to the role of college students’ role in spreading COVID-19 [22]. Indeed, high-density structures in university environments like dorms, classrooms, gyms, and socializing avenues (e.g., bars, house parties) can be key areas where close face-to-face interaction occurs and, thus, propagates COVID-19.

The inclusion criteria required that students must be enrolled in participating classes where they are required to participate in at least two studies during that semester, which counts for 5% of their total grade. Alternative assignments were available if the students declined to consent in any of the available studies in the research pool. The selected university still had the majority of its classes face-to-face, and the county had minimal restrictions regarding business openings (e.g., restaurants capped at 50%, bars to close at midnight, masks required).

The study launched on 18 September 2020 and concluded on 16 November 2020. The reason for terminating the study on 16 November 2020 was that the state and county were soon to enact new COVID-19 restrictions, which may have affected the rates of interpersonal contact and places visited. After removing 18 cases of missing data (i.e., more than half of items were missing), the final sample size was 397: 138 in the baseline condition, 127 in the retrieval cue condition, and 132 in the context-based condition. The participants’ ages ranged from 18 to 42, with an average age of 19.94 (SD = 2.35). Regarding gender, 262 (65.8%) identified as woman, 135 (33.9%) as men, and one (0.03%) preferred not to disclose. Finally, 305 (76.6%) identified as white, 50 (12.6%) as African American, 17 (4.30%) as Hispanic, 10 (2.50%) as Asian, 1 (0.03%) as Native Hawaiian or other Pacific Islander, and 15 (3.80%) as Other (see Table 1).

### 2.3. Apparatus

The experiment was conducted online using Qualtrics. Participants were required to complete the study over a computer and laptop in the setting of their choice (e.g., at home). Students were not able to participate using a mobile device. Our decision to disallow mobile devices stemmed from the required textual input needed for providing contacts and places visited, which may be more difficult to supply over a mobile device.

### 2.4. Procedure

Participants took part in the experiment individually over a computer at the setting of their choice. When a student logged into the research participation pool website, there were a variety of studies they could choose to complete. The research pool website included the name of the study, which read “Mock Contact Tracing Study”. A brief description of the study below the title included the text, “This study will ask if you can remember, using anonymous pseudonyms, the people and places you have visited from the last four days”.

If a student clicked to participate in the study, they first encountered a consent form, which described the purpose, description, and potential benefits of the survey. A key element stressed was that participation was anonymous and when entering contacts and places visited, that the participants should use pseudonyms or nicknames because we were not interested in the actual contacts or places visited, just that if they could remember them. Examples of pseudonyms were given as helpful suggestions (e.g., my best friend, my favorite bar).

The experiment began with definition of “contacts” and “places”. Contacts were defined as “being in the same proximity within six feet with anybody for at least a duration of 10 min”, a definition borrowed from John Hopkins University [23]. An illustration of two people within a six feet radius was presented to give an example. A place was defined as a “populated location visited outside your home. Examples include “grocery store, doctor’s office, work building, restaurant, night club, etc.” After students agreed upon the definitions of contact and places, they were randomized into one of the three independent conditions (see “Measures” for a description) for eliciting contacts and places. After reporting contacts and places, participants were all given items for control variables. These items included beliefs regarding COVID-19, mood, well-being, ability, and employment.

### 2.5. Measures

#### 2.5.1. Contact Elicitation

Contact elicitation was the key manipulation. There were three independent conditions. In the baseline approach, participants were asked to report their close social contacts motivated by wording from the General Social Survey [24]. The question was adapted to reflect places visited as well. The “places” question read as follows:

Please take several minutes to think back from the **last four days.** Can you recall all of the **places** you have visited during those 4 days? Please use pseudonyms as we are not interested in the actual place.

The “contact” question similarly read as follows:

Please take a few minutes to think back from the **last four days.** Can you recall all of the **contacts** you have made during those 4 days? If so, enter each name below. Please use pseudonyms as we are not interested in their real names.

For the retrieval cue approach, we followed procedures laid out by Hiesh [10]. Before eliciting contacts, participants were instructed to look at their (1) cellphone calls and texts from the last four days, (2) last 30 emails, and (3) recent posts, tags, and messages they had sent or received on their preferred social media (e.g., Facebook, Twitter, etc.) as retrieval cues. Then, the same two questions above were provided to elicit contacts and places.

Finally, for the context-based approach, four activity foci were activated to elicit contacts and places: (1) homelife, (2) worklife and resources, (3) leisure, and (4) other. More specifically, the questions read as follows:**Homelife**: Many people have permanent residencies where they live with family members, partners, and/or friends.Can you identify all of the **homes** you have visited within the **past 4 days**?Can you identify all of the people that you have had any contact regarding **homelife** within the **past 4 days**? These include people who you live with and people who have visited your home. These also include people whose home you have visited.**Worklife and resources**: Many people must travel to places to gather resources for daily life. These include things like their workplace and shopping for food, clothes, gas, medical prescriptions, home supplies, or various technologies (e.g., a new phone).Can you identify all of the places that you have visited regarding **worklife/resources** within the past 4 days?Can you identify all of the people that you have had any contact regarding **worklife/resources** within the **past 4 days**? These include people who you work closely with and people you interact with to get the resources you need.**Leisure**: Many people engage in leisure activities with other people outside of work. These include things like going to churches, social clubs, restaurants, bars, gyms, going to the beach, and public events.Can you identify all of the places that you have visited regarding **leisure** within the **past 4 days**?Can you identify all of the people that you have had any contact regarding **leisure** within the **past 4 days**? These include people who you engage with in leisure activities outside of work.**Other**: Are there any other places you have visited that you have not entered yet?Can you identify all of the other places that you have visited within the **past 4 days**?Can you identify more contacts within the **past 4 days**?

In all three conditions, participants provided contacts and places in a text box for convenience. Each participants’ text was counted individually by the authors to check for duplicates (e.g., a participant lists the same contact or place twice) and general adherent to the survey procedures (e.g., removing participants that did not follow instructions). The final two dependent variables were the amount of unique contacts and places elicited by each participant.

#### 2.5.2. Control Variables

Past research on individual level factors and network alter recall can help give guidance for additional control variables to include. Our review identified such factors as gender [25], age [10], race [10], mood [12], well-being [13], and physical functioning [26]. Standard items were used to gather information on demographics like gender, age, and race. For mood (e.g., I am generally in a happy mood) we included nine items (α = 0.91) from a scale developed by [27]. For well-being (e.g., How satisfied are you with your current well-being?), we used the nine items (α = 0.86) provided by Shea et al. [13]. Physical functioning (e.g., Do you have difficulty walking?) was measuring using five items (α = 0.81) from Ross and Mirowsky [28].

Finally, we sought to include three single item questions regarding attitudes to COVID-19 and masks. It may be the case that individuals who have more relaxed views on the risk of COVID-19 and the efficacy of masks may be more likely to engage in interpersonal contacts and visit more places. The questions included were “On a scale of 1-10, how serious of a public health threat is COVID-19?”, “How worried are you about getting COVID-19?” and “Do you think wearing a mask helps to reduce the spread of COVID-19?” All items are listed in the Appendix A for reference.

### 2.6. Analysis

Because both dependent variables represent “count” data (i.e., contacts and places visited) with skewed distributions, we opted for a negative binomial regression model. A negative binomial regression was chosen over a Poisson model particularly because the variance was greater than the mean for both contacts and places [29]. Typical assumptions of independence, multicollinearity, and linearity were checked and satisfied. Robust standardized errors were used for more precise estimates of coefficient values.

Effect size measures were calculated following Coxe’s [30] method for calculating standardized mean differences (SMD): 0.20–0.50, small; 0.50–0.80, medium; and above 0.80, large. In step 1, only the key contact elicitation variable was entered. In step 2, the control variables were added to see if they impacted the association between method of contact elicitation and the two dependent variables. All analysis was performed using Stata 16 [31].

## 3. Results

On average, the baseline approach elicited 9.28 contacts (SD = 8.69), the retrieval cue elicited 10.61 (SD = 7.68), and the context-approach elicited 13.13 (SD = 7.60). The number of places visited was lower. On average, the baseline approach elicited 5.72 places (SD = 2.30), the retrieval cue elicited 5.67 (SD = 3.08), and the context-approach elicited 8.27 (SD = 4.01).

H1 predicted that the retrieval cue approach, which instructs contacts to view their previous text messages, emails, and social media posts, would stimulate higher recall of contacts and places compared to a baseline approach (see Table 2). H1 was not supported when predicting contacts (*Est.* = 0.13, *SE* = 0.10, *p* = 0.13, *SMD* = 0.22) or places (*Est.* = −0.03, *SE* = 0.05, *p* = 0.47, *SMD* = −0.08). As such, the null hypothesis assuming no differences between the baseline approach and retrieval cue was not rejected.

H2 predicted that the context-based approach, which primes participants to think about contacts and places with respect to homelife, worklife/resources, leisure, and other, would elicit more contacts and places than a baseline approach. H2 was supported. Compared to the baseline approach, the context-based approach garnered medium effect sizes when predicting contacts (*Est.* = 0.34, *SE* = 0.09, *p* < 0.01, *SMD*= 0.62) and large effect sizes when predicting places (*Est.* = 0.33, *SE* = 0.05, *p* < 0.01, *SMD*= 0.87).

RQ1 asked about differences between the retrieval cue approach and the context-based approach. To keep the effects of the control variables, coefficient contrasts were estimated using Bonferroni corrected z-scores with post-estimation (see Figure 1). In terms of contacts, the coefficient contrast (ϛ) between the retrieval cue and context-approach was significant (ϛ = 0.21, *SE* = 0.08, *z* = 2.54, *p* = 0.03). With respect to places, the coefficient contrast was significant as well and larger (ϛ = 0.38, *SE* = 0.06, *z* = 6.26, *p* < 0.01).

Finally, although the addition of the control variables did not impact any of the effects of the experimental manipulations, some of the results were worth noting. Demographically, participants who identified as women were likely to report more contacts (*Est.* = 0.16, *SE* = 0.07, *p* = 0.03), as well as those who identified as White when compared to those is identified as Asian, African American, and Hawaiian/Pacific Islander. Psychologically, those in better moods reported more contacts (*Est.* = 0.12, *SE* = 0.06, *p* = 0.04), and those who were less worried about COVID-19 reported fewer contacts (*Est.* = −0.08, *SE* = 0.08, *p* = 0.04) and places (*Est.* = −0.04, *SE* = 0.02, *p* = 0.10).

## 4. Discussion

Valente [32] defines social network interventions as consisting of “behavior change programs that use social network data to identify specific people or groups to deliver and/or receive the behavior change program” (p. 146) and summarized six types of interventions. Contact tracing largely falls under the realm of network recruitment, where an ego’s alters (i.e., contacts) are identified and recruited to quarantine and get tested for a possible infectious disease. Given the large costs associated with face-to-face interviews to conduct contact tracing, we sought to contribute to innovations in social network data collection methods for health-based interventions by developing a survey-based contact tracing instrument.

More specifically, the purpose of this research was to test the effectiveness of three different egocentric survey methods for eliciting contact tracing data: a (1) baseline, (2) retrieval cue, and (3) context-based approach. Overall, significantly more contacts and places were elicited when participants were randomized into the context-based condition, which primed users on activity foci of homelife, worklife/resources, leisure, and other. Theoretically, the results provide support field theory’s suggestion that better recall can be stimulated when relevant activity foci are triggered. Practically, the study begins to shed light on more effective ways to design egocentric contact tracing surveys. Each of these realms are discussed further below.

There are a few examples of research that test somewhat competing methods of eliciting interaction-based contacts. The current study nearly replicated Hsieh’s [10] study, which found that a retrieval cue approach elicits more contacts than a baseline approach (*p* = 0.13). However, a few caveats much be mentioned before any premature conclusions. First, Hsieh [10] elicited affective contacts (i.e., contacts whom the ego discusses important matters with) not interaction contacts as compared to the current study. Second, there could be selection bias in that the current study only looked at college students from a single university. Because different age groups and generations tend to use technology in different ways and have different attitudes on such technology (e.g., [33]), it may be the case that the retrieval cue approach may not be as effective with younger participants. Future research may be served well by including a more generalizable sample with larger diversity of age groups, locations, and socio-economic status. Likewise, because of the convenient sampling strategy used, we cannot exclude that there may be some levels of volunteer bias, meaning that those who chose to participate in the experiment differ in some characteristics than those who chose not to. Future research may consider more probability based methods of sampling to rule out such biases.

Nevertheless, the results found that the context-based approach elicited more contacts and places visited than the baseline and retrieval cue approaches. Following field theory, Corman and Scott [34] theorize a similar network reticulation theory for explaining the relationship between perceived and observable networks (see also [35]). Again, the key here is to activate relevant activity foci to stimulate network recall of observable interactions. However, a key limitation for adapting the context-based approach of Bidart and Charbonneau [17] is determining relevant activity foci. Similar survey adaptations of the approach are typically contextual, including foci relevant to education paths, youth foster care, and individuals who live alone (see [17], fn. 7). The four foci here are by no means a final set and will need to be continually refined in order to create a gold-standard survey to elicit alters for contact tracing. Finally, just because an instrument elicits more contacts and places, this does not necessarily mean it performs the same as a face-to-face interview. Further studies might consider comparing survey-based approaches to more intensive qualitative approaches to really see how much they differ.

## 5. Conclusions

The main innovation the current research makes is with respect to developing changes in social network data collection methods for one of the oldest types of relational health-based interventions: contact tracing. This is the first study, to the best of our knowledge, that tests different methods for eliciting interaction-based contacts and places. Overall, the current results provide support for the context-based approach, which first attempts to trigger relevant activity foci and then elicits contacts corresponding to such foci. Participants in the context-based were significantly more likely to elicit more contacts and places visited compared to traditional baseline approaches and the retrieval cue approach. The identification of more true positive contacts and places may allow public health officials to better implement quarantine and testing interventions because fewer contacts and places are slipping through the cracks.

To put it simply, some communicative interactions may appear on the surface as random (e.g., a conversation with a bartender), which may make it difficult for individuals to remember them. However, this only appears to be the case if they are not viewed as situated action and stripped away from relevant activity foci (e.g., leisure → going to a bar). If it is true that contact tracing is only as good as the data that can be extracted, then making strides at developing a reliable and valid survey for easier and faster deployment can significantly impact public health with respect to infectious diseases that diffuse through interactive-based contact.

## Figures and Tables

**Figure 1 ijerph-18-01466-f001:**
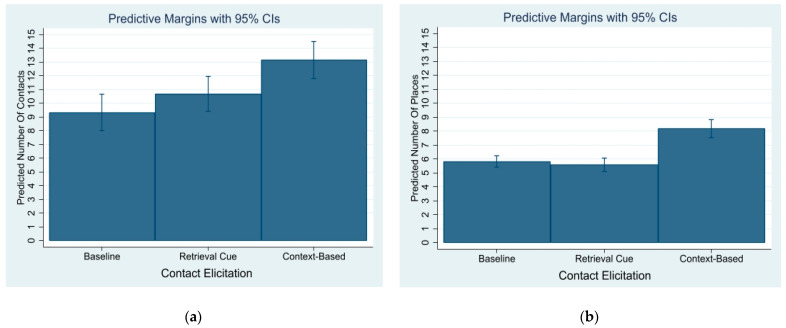
(**a**) Plots of post-estimation mean differences for contacts. (**b**) Plots of post-estimation mean differences for places.

**Table 1 ijerph-18-01466-t001:** Correlations and descriptive statistics.

	(1)	(2)	(3)	(4)	(5)	(6)	(7)	(8)	(9)	(10)
Contacts (1)										
Places (2)	0.42									
Gender (3)	0.07	0.18								
Age (4)	0.04	0.01	−0.05							
Mood (5)	0.09	0.07	−0.04	0.01						
Well-being (6)	0.09	0.08	−0.04	−0.04	0.62					
Physical functioning (7)	−0.06	−0.10	0.14	−0.05	−0.06	−0.10				
Mask efficacy (8)	−0.11	−0.09	0.06	0.03	−0.04	−0.09	−0.01			
COVID-worry (9)	−0.15	−0.12	0.06	−0.01	−0.12	−0.17	0.09	0.39		
COVID-serious (10)	−0.13	−0.08	0.20	0.05	−0.05	−0.12	0.02	0.52	0.49	
Mean	10.98	6.55	1.66	19.94	3.83	3.84	1.05	4.26	2.22	7.17
Median	9.00	6.00	2.00	20.00	3.88	3.88	1.00	4.00	2.00	7.00
SD	8.16	3.41	0.47	2.35	0.63	0.64	0.19	0.85	0.98	2.24

**Table 2 ijerph-18-01466-t002:** Robust negative binomial regression.

**Contacts as Dependent Variable**						
**Variable**	**Coef. (SE)**	***p***	***SMD***	**Coef. (SE)**	***p***	***SMD***
H1: Retrieval Cue	0.13 (.10)	0.18	0.22	0.13 (0.09)	0.13	0.22
H2: Context-Based	0.34 ** (0.09)	0.00	0.62 *	0.34 ** (0.09)	0.00	0.62 *
Control Variables Gender				0.16 * (0.07)	0.03	
Age				0.01 (0.01)	0.53	
Asian				−0.33 * (0.15)	0.03	
African American				−0.34 ** (0.10)	0.00	
Hispanic				0.07 (0.15)	0.63	
Hawaiian/Pacific Islander				−0.65 ** (0.10)	0.00	
Other Race				0.24 † (0.15)	0.10	
Mood				0.12 * (0.06)	0.04	
Well-being				−0.02 (0.06)	0.64	
Physical functioning				−0.13 (0.20)	0.52	
Mask efficacy				−0.04 (0.04)	0.34	
COVID-worry				−0.08 * (0.04)	0.03	
COVID-seriousness				−0.01 (0.02)	0.56	
Intercept	2.23 ** (0.08)	0.00		2.04 ** (0.51)	0.00	
**Places as Dependent Variable**						
**Variable**	**Coef. (SE)**	***p***	***SMD***	**Coef. (SE)**	***p***	***SMD***
H1: Retrieval Cue	−0.01 (0.05)	0.86	−0.02	−0.03 (0.05)	0.47	−0.08
H2: Context-Based	0.36 ** (0.05)	00	0.89 *	0.33 ** (0.05)	0.00	0.87 *
Control variables Gender				0.22 ** (0.05)	0.00	
Age				−0.01 (0.01)	0.90	
Asian				0.17 (0.06)	0.14	
African American				−0.19 ** (0.06)	0.00	
Hispanic				0.06 (0.15)	0.66	
Hawaiian/Pacific Islander				0.06 (0.15)	0.66	
Other Race				−0.08 (0.09)	0.39	
Mood				0.05 (0.04)	0.20	
Well-being				−0.01 (0.05)	0.83	
Physical functioning				−0.30 * (0.14)	0.03	
Mask efficacy				−0.02 (0.03)	0.55	
COVID-worry				−0.04 † (0.02)	0.10	
COVID-seriousness				−0.01 (0.01)	0.75	
Intercept	1.74 ** (0.03)	0.00		1.80 ** (0.35)	0.00	

Note: ** *p* ≤ 0.01, * *p* ≤ 0.05, † ≤ 0.10. SMD effect sizes: 0.20–0.50 = small; 0.50–0.80 = medium; and above 0.80 = large. For Gender, 1 = Male, 2 = Woman. For race, the reference group is White.

## Data Availability

Anonymous link to dataset: https://osf.io/u7bda/?view_only=0394755b34a843808ac5cf34e8369b11.

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
