# Peer review of "An Egocentric Network Contact Tracing Experiment: Testing Different Procedures to Elicit Contacts and Places"

_ijerph, 2021, doi:10.3390/ijerph18041466_

Round 1

Reviewer 1 Report

I have read the paper entitled "An Egocentric Network Contact Tracing Experiment". In that work, the authors compare various egocentric survey methods for extracting contract tracing data, namely, a baseline approach, a trtrieval cue approach and a contect-based approach. To that end, they follow a statistical methodology. According to the results reported in this work, participants in the context-based condition were significantly more likely recall more contacts and places than the other two conditions. MAJOR COMMENTS I have various comments on this manuscript. To start with, the results are not particularly ground-breaking and the study is such a straightforward application of approaches in the theory of social networks. However, the final outcome may be of interest to practitioners in the area. My suggestions are the following in order to improve the manuscript. 1. Appendix A must be moved to the 'Supplementary Materials'. 2. The manuscript must be shortened in order to account for the changes in Comment 1. 3. The authors mus discuss the sampling scheme. Which type of sampling approach was chosen? Which were the reasons to choose such approach? 4. In order to carry out your hypotheses testing, which assumptions did you have to make? How can you be sure that they are satisfied? 5. Samples were taken from a "southern University". In light of that constraint, how can be expect to generalize the scope of the results? The conclusion is actually of local character? Or can we extend it to populations outside that "southern University"? Why? In my opinion, this work is a short note, not a full research article. I expect that the author comply with my suggestions and respond my questions convincingly.

Author Response

R1 Response

Thank you for the review of the current manuscript. We will try to address your comments on how to improve the manuscript below.

COMMENT #1: “Appendix A must be moved to the ‘Supplementary Materials’

RESPONSE #1: “Appendix A has now been moved the ‘Supplementary Materials’ section

COMMENT #2: “The manuscript must now be shortened in order to account for the changes in Comment 1”

RESPONSE #2: The manuscript has been changed from 6099 words to 4791 words (main body text)

COMMENT #3: “The authors must discuss the sampling scheme. Which type of sampling approach was chosen? Which were the reasons to choose such an approach?

RESPONSE #3: We have made it more clear that we used convenience sampling from a research participation pool in Section 2.2:

“To recruit participants for the experiment, we used convenience sampling from a College-wide research participation pool from a large southern University.”

In the limitations, we know acknowledge that like all convenient samples, there may be volunteer bias (ln 338):

“Likewise, because of the convenient sampling strategy used, we cannot exclude that there may be some levels of volunteer bias, meaning that those who chose to participate in the experiment differ in some characteristics than those who chose not to. Future research may consider more probability based methods of sampling to rule out such biases.”

COMMENT #4: “In order to carry out your hypotheses testing, which assumptions did you have to make? How can you be sure they are satisfied?”

RESPONSE #4: Because we use a generalized linear model (negative binomial regression), we checked for basic assumptions like

  • Independence: We ran Durbin-Watson autocorrelation tests for contacts (DW = 2.00) and places (DW = 2.251). Because these values were between 1 and 3 (Field, 2017), we deemed that independence was satisfied.
  • Multicollinearity: We checked to see if there were excessive correlations between predictor variables. None of the Variance Inflation Factors (VIFs) were greater than 2 and none of the tolerance values were below 0.10, which suggests no apparent multicollinearity.
  • Linearity: We looked at the scatterplots of the standardized residuals against the predicted values. No systematic non-linear relationship was evident.

The assumption of normality was not met with the current ‘count’ data, which is why we opted for a negative binomial regression model. These assumptions have been described in text now (ln 267-269):

“Because both dependent variables represent ‘count’ data (i.e., contacts and places visited) with skewed distributions, we opted for a negative binomial regression model. A negative binomial regression was chosen over a Poisson model particularly because the variance was greater than the mean for both contacts and places [28]. Typical assumptions of independence, multicollinearity, and linearity were checked and satisfied.”

Field, A. (2017). Discovering statistics using IBM SPSS statistics. Thousand Oaks, CA: Sage.

COMMENT #5: “Samples were taken from a “southern University” Why? In my opinion, this work is a short note, not a full research article. I expect that the author comply with my suggestions and respond my questions convincingly.

RESPONSE #5: There are two reasons for the sample selection. First, different counties have different COVID-19 policies. For instance, New York City is more restrictive than Southern Indiana. The type of restriction in place will surely influence the amount of contacts people make and places visited. For instance, if a populated place is closed for business (e.g., the local pub), people won’t report going to it. As such, we aimed for a sample where the local COVID-19 restrictions were constant. An alternative would have been to garner a larger, more representative sample. However, the funding allocated to this study prevented us from doing so. We make that point and expand on it on ln 142-145:

“To recruit participants for the experiment, we used convenience sampling from a College-wide research participation pool from a large southern University. A University sample was chosen for two reasons. First, COVID-19 restrictions (e.g., closed businesses, stay-at-home orders, etc.) differ state-by-state. In order to control for policy differences, we opted for keep geographic location constant.”

The second reason is the people in the sample itself: college students. With respect to COVID-19, college students have been a focal interest among spreading COVID-19 because universities lend themselves to high-density interactive structures like classrooms, dorms, gyms, libraries, and various socializing avenues (house parties, bars, etc.). Indeed, various research from the NY Times has shown that risk of contracting COVID-19 is significantly higher in college towns, especially since the Fall 2021 semester begin. As such, if college students are part of a group that seems to be propagating the virus, we think they are a justified group to sample. We know make that point a little clearer (ln 146-149):

“Second, there is an increasing attention paid to the role of college students’ role in spreading COVID-19 [21]. Indeed, high-density structures in university environments like dorms, classrooms, gyms, and socializing avenues (e.g., bars, house parties) can be key areas where close face-to-face interaction occurs and thus, propagate COVID-19.”

NY Times References

https://www.nytimes.com/2020/12/12/us/covid-colleges-nursing-homes.html

https://www.nytimes.com/2020/09/06/us/colleges-coronavirus-students.html

Reviewer 2 Report

This is a well-written and timely paper.  It provides important information and insight into ways to address the current pandemic.  I only had a few comments, and these are based on the fact that students of public health (or students in general) will most likely be reading this.  You have provided excellent context, but there are a few places where just a little more explanation would be helpful to the reader.

line 15: "...were prompted to anonymously provide..."  add that you prompted them in a way that was appropriate to the treatment they were assigned to.

I wondered the impact of the controls.  Could you add more about that?

line 328 "...does not necessarily means it performs..."  should be "...mean..."

Author Response

R2 Response

Thank you for the review of the current manuscript. We will try to address your comments on how to improve the manuscript below.

COMMENT #1: line 15“were prompted to anonymously provide’…add that you prompted them in a way that was appropriate to the treatment they were assigned to

RESPONSE #1: Good point here. We have changed that sentence a little bit to make it clearer. It now reads:

“A sample of 397 college students were randomized into one condition each. They were prompted to anonymously provide contacts and populated places visited from the past four days depending on what condition they were given.”

COMMENT #2: “I wondered about the impact of the controls. Could you add more about that”?

RESPONSE #2: Yes! We have added a last paragraph to the results that briefly summarized the results of the control variables (ln 301-307):

“Finally, although the addition of the control variables did not impact any of the effects of the experimental manipulations, some of the results were are worth noting. Demographically, participants who identified as women were likely to report more contacts (Est. = 0.16, SE = 0.07, p = 0.03), as well as those who identified as White when compared to those is identified as Asian, African American, and Hawaiian/Pacific Islander. Psychologically, those in better moods reported more contacts (Est. = 0.12, SE = 0.06, p = 0.04) and those who were less worried about COVID-19 reported more contacts (Est. = -0.08, SE = 0.08, p = 0.04) and places (Est. = -0.04, SE = 0.02, p = 0.10).”

COMMENT #3: line 328 “…does not necessarily means it performs...” should be “…mean…”

RESPONSE #3: We have added this change from “means” to “mean”

Reviewer 3 Report

Since some infectious diseases like COVID-19 amass at such a great scope that traditional methods of conducting contact tracing remain difficult to implement, this paper tests the effectiveness of three different egocentric survey methods for extracting contact tracing data including a baseline approach, a retrieval cue approach and a context-based approach. the research contributes to developing innovative social network data collection methods for contract tracing survey instruments. The results provide support for the context-based approach which first attempts to trigger relevant activity foci and then elicits contacts corresponding to such foci. As for the research content of this paper, I put forward relevant questions and suggestions for improvement as follows:
1.The title of this paper is “An Egocentric Network Contact Tracing Experiment”, which could barely reflect the specific research content and research background of this paper. It is suggested that the author modify the title of this paper to make it more specific
2. Compared with the existing research, what is the main innovation of this paper? Please clarify it in detail.
3. Is the sample size of the experiment sufficient enough to guarantee the accuracy of the research results?
4. In this paper, the author states “contacts were defined as ‘being in the same proximity within six feet with anybody for at least a duration of 10 minutes’”. How does the author make such a definition?
5. What is the most significant difference among the three egocentric survey methods, which are(1) a baseline approach, (2) a retrieval cue approach, and (3) a context-based approach?
6. What is the practical meaning of this paper to quarantine measures?

Author Response

Reviewer 3

Thank you for the review of the current manuscript. We will try to address your comments on how to improve the manuscript below.

COMMENT #1: The title of this paper is “An Egocentric Network Contact Tracing Experiment”, which could barely reflect the specific research content and research background of this paper. It is suggested that the author modify the title of this paper to make it more specific

RESPONSE #1: You are right; the title is a bit vague. We have added a second part to it. It is now titled:

“An Egocentric Network Contact Tracing Experiment: Testing Different Procedures to Elicit Contacts and Places”

COMMENT #2: Compared with the existing research, what is the main innovation of this paper? Please clarify it in detail.

RESPONSE #2: Thanks for giving us a chance to offer a big-picture take-away. Essentially, this is the first study to test the effectiveness of contact and place elicitation instruments. Some research, like (Hsieh, 2015), testing the effectiveness of strong ties, but not interaction ties, which is much more relevant for contact tracing. Put simply, there is a lot of contact tracing being done, but there really hasn’t been much testing on the reliability and validity of the methods being used to elicit the data necessary for contact tracing. To us, that is the main innovation of the current study. We briefly summarize that point in lines 356-359:

“The main innovation the current research makes is with respect to developing changes in social network data collection methods for one of the oldest type of relational health-based interventions: contact tracing. This is the first study, to the best of our knowledge, that tests different methods for eliciting interaction-based contacts and places.”

COMMENT #3: Is the sample size of the experiment sufficient enough to guarantee the accuracy of the research results?

RESPONSE #3: According to traditional a-priori power, a sample of 159 subjects are needed to detect at least medium-sized effects. Because the current sample of 397 exceeds this number, we are confident we can interpret medium and large sized effects, but not small effect sizes. We now more clearly report the results of the power analysis in lines 139-140:

“An a priori power analysis with the program GPower [21] suggests that at least 159 participants are needed to detect medium-sized effects (1 – β = 0.80, f = 0.25, α = 0.05).”

COMMENT #4: In this paper, the author states “contacts were defined as ‘being in the same proximity within six feet with anybody for at least a duration of 10 minutes’”. How does the author make such a definition?

RESPONSE #4: This definition was taken from John Hopkins University’s course on contact tracing (Gurley, 2020) one of the authors enrolled in. We have now cited the resource (ln 186-188):

“The experiment began with definition of “contacts” and “places”. Contacts were defined as “being in the same proximity within six feet with anybody for at least a duration of 10 minutes”, a definition borrowed from John Hopkins University [21].”

COMMENT #5: What is the most significant difference among the three egocentric survey methods, which are (1) a baseline approach, (2) a retrieval cue approach, and (3) a context-based approach?

RESPONSE #5: There is a couple different ways to answer this question, so we will try to answer in both ways. Conceptually, the difference is in how you ask the questions. Section 2.5.1, entitled “Contact Elicitation” describes the three experimental manipulations based on these three approaches. The Baseline approach is quite general:

“In the baseline approach, participants were asked to report their close social contacts motivated by wording from the General Social Survey [22]. The question was adapted to reflect places visited as well. The “places” question read:

Please take several minutes to think back from the last four days. Can you recall all of the places you have visited during those 4 days? Please use pseudonyms as we are not interested in the actual place.

The “contact” question similarly read:

Please take a few minutes to think back from the last four days. Can you recall all of the contacts you have made during those 4 days? If so, enter each name below. Please use pseudonyms as we are not interested in their real names.”

The retrieval cue approach, has the same questions, except that it asks the participant to use various retrieval cues:

“Before eliciting contacts, participants were instructed to look at their (1) cellphone calls and texts from the last four days, (2) last 30 emails, and (3) recent posts, tags, and messages they have sent or received on their preferred social media (e.g., Facebook, Twitter, etc.) as retrieval cues. Then, the same two questions above were provided to elicit contacts and places.”

Finally, the context-based approach asks about contacts and places with respect to (1) homelife, (2) worklife & resources, (3) leisure, and (4) other. We have copied the exact language of the survey used in the experiment so that the reader is clear on what language was used.

Moreover, if the question is more to do with statistical differences, we report the average values for contacts and places in the first line of the Results section (ln 276-279):

“On average, the baseline approach elicited 9.28 contacts (SD = 8.69), the retrieval cue elicited 10.61 (SD = 7.68), and the context-approach elicited 13.13 (SD = 7.60). The number of places visited was lower. On average, the baseline approach elicited 5.72 places (SD = 2.30), the retrieval cue elicited 5.67 (SD = 3.08), and the context-approach elicited 8.27 (SD = 4.01).”

COMMENT #6 What is the practical meaning of this paper to quarantine measures?

RESPONSE #6: Interesting question here. For quarantining, the results suggest that more ‘true positives’ would hypothetically be asked to get tested and quarantined if a more context-based approach is used. Likewise, for places, this would allow contact tracers to more accurately identify ‘hot-spots’ and ‘super-spreader events’ if more ‘true positive’ places are elicited.

We have modified the Introduction just a bit to include quarantine implications (ln 39):

“Indeed, the effectiveness of contact tracing is only as good as the data that can be extracted. If ideal standards are not followed for collecting contact tracing data, then most efforts to take advantage of this data will likely fall short. Potential at-risk contacts may never be discovered for treatment/quarantine and they could continue to propagate COVID-19. Likewise, hotspots and clusters are likely not to be discovered, preventing the public from being informed to follow social distancing.”

We also now briefly mention such practical meanings in the Conclusion (ln 361):

“Participants in the context-based were significantly more likely to elicit more contacts and places visited compared to traditional baseline approaches and the retrieval cue approach. The identification of more true positive contacts and places may allow public health officials to better implement quarantine and testing interventions because less contacts and places are slipping through the cracks.”

Round 2

Reviewer 1 Report

The authors have responded satisfactorily my criticisms.

Author Response

Thank you for the feedback!

Reviewer 3 Report

I think the current version is acceptable.

Author Response

Thank you for the feedback!